# HIV Infection and Spread between Th_17_ Cells

**DOI:** 10.3390/v14020404

**Published:** 2022-02-16

**Authors:** Janet P. Zayas, João I. Mamede

**Affiliations:** Department of Microbial Pathogens and Immunity, Rush University Medical Center, Chicago, IL 60612, USA; janet_p_zayas@rush.edu

**Keywords:** HIV-1, CD4^+^ T cells, Th_17_ cells, pathogenesis, reservoirs, cell-to-cell spread, cell-free spread

## Abstract

HIV mainly targets CD4^+^ T cells, from which Th_17_ cells represent a major cell type, permissive, and are capable of supporting intracellular replication at mucosal sites. Th_17_ cells possess well-described dual roles, while being central to maintaining gut integrity, these may induce inflammation and contribute to autoimmune disorders; however, Th_17_ cells’ antiviral function in HIV infection is not completely understood. Th_17_ cells are star players to HIV-1 pathogenesis and a potential target to prevent or decrease HIV transmission. HIV-1 can be spread among permissive cells via direct cell-to-cell and/or cell-free infection. The debate on which mode of transmission is more efficient is still ongoing without a concrete conclusion yet. Most assessments of virus transmission analyzing either cell-to-cell or cell-free modes use in vitro systems; however, the actual interactions and conditions in vivo are not fully understood. The fact that infected breast milk, semen, and vaginal secretions contain a mix of both cell-free viral particles and infected cells presents an argument for the probability of HIV taking advantage of both modes of transmission to spread. Here, we review important insights and recent findings about the role of Th_17_ cells during HIV pathogenesis in mucosal surfaces, and the mechanisms of HIV-1 infection spread among T cells in tissues.

## 1. Introduction

Human immunodeficiency virus (HIV) infection is a global public health concern and the etiological agent of acquired immune deficiency syndrome (AIDS). There is no known cure for HIV infection to date, and the demand for an HIV cure is on the rise, given its associated costs, adverse events, stigma, as well as lifelong treatments using antiretroviral therapy (ART) [1]. Worldwide, about 37.7 million people were living with HIV by the end of 2020, with an estimated 27.5 million people able to access ART during the same year [2,3,4]. Although new HIV infections are decreasing globally, a devastating estimated 1.5 million people became newly infected in 2020 [4].

The exact HIV pandemic origin is unknown; however, it is well documented that HIV infection spread from non-human primates to humans around the 1900s [5,6]. HIV is a complex retrovirus of the lentivirus family [7]. Retroviruses are characterized by the fact that they carry their genetic material as single-stranded RNA (ssRNA) along with the necessary components to hijack the genetic machinery of a target cell to replicate itself. The main types of HIV infection include HIV-1 and HIV-2, which present differences in replication and pathogenicity but share similarities in genetic and biological properties [8]. Most HIV infections correspond to HIV-1 which is also documented to be more pathogenic than HIV-2 [8,9]. If left untreated, HIV-1’s mortality rate is over 95% [8,9]. HIV-1 is an enveloped retrovirus with two copies of an ssRNA genome, which enters the host primarily through mucosal surfaces, where it replicates after the integration of the newly retro-transcribed double-stranded DNA (dsDNA) [5,10,11]. Predominant routes of HIV-1 transmission include sexual contact, percutaneous (through contaminated needles and blood products), and perinatal [5,10,12]. Worldwide, about 90% of new HIV infections are attributed to sexual transmission, with most infected people being women [13,14,15].

One of the hallmarks of HIV-1 is that it selectively infects and depletes CD4^+^ T cells, disrupting T cell homeostasis [8,16]. As a result, HIV patients develop a rapid drop in T cell counts, a ramp-up phase of viremia, and impaired immunity [8,16]. The problem is complex, since as soon as HIV-1 infection initiates, reservoirs may also establish within resting memory CD4^+^ T cells and other cells [17]. HIV is well-versed in hiding from the immune system and persisting regardless of a lifetime under suppressive ART [17,18]. ART can inhibit new rounds of viral replication, reduce plasma viral load below clinical detection limits (20 to 50 RNA copies/mL), and interrupt disease progression [17,18]; however, some HIV-infected cells survive as persistent and latently infected cells [19]. Although early initiation of ART has brought improved life expectancy to patients, it has its limitations, and the different treatment regimens are unable to deplete latent reservoirs, prevent infection establishment, and efficiently suppress new infections [16,18]. HIV-1 latent reservoirs persist undetectable regardless of ART, annihilating all chances for ART to eradicate HIV infection. Current proposed strategies to eliminate latent HIV-1 reservoirs include “shock and kill”, “block and lock”, and “lock in and apoptosis” [17]. The “shock and kill” approach aims to “shock” or reactivate the latent virus using latency-reversing agents (LRAs) such as toll-like receptor (TLR) agonists and histone deacetylases (HDACs), and then “kill” infected cells or eliminate the latent reservoir via targeted cytotoxic T lymphocyte (CTL) response [17,20]. The “block and lock” approach aims to enhance the latent virus state by “blocking” HIV transcription and “locking” the HIV promoter in a deep or super latent state using small interfering RNAs (siRNAs) or trans-activator of transcription (Tat) inhibitors to disrupt epigenetic regulators or viral replication [17,20]. Moreover, the “lock in and apoptosis” approach aims to block virus budding from the cell using LRAs, and Pr55^Gag^ HIV-1 protease inhibitor [17]. Alternative approaches to target latent HIV reservoirs involve gene therapy via stem cell transplantation or via gene editing using CRISPR/Cas9 and zinc-finger nucleases (ZFN) [17,20]. Still, an effective strategy to eliminate HIV infection remains elusive.

Remarkable research efforts have led to a better understanding of HIV target cells (mainly CD4^+^ T cells), and HIV cellular reservoirs, including naïve CD4^+^ T cells (TN), stem cell-like memory (TSCM), central memory (TCM), transitional memory (TTM), effector memory CD4^+^ T cells (TEM), T helper 1, 2, 17, 9 (Th_1_, Th_2_, Th_17_, Th_9_) cells, regulatory T cells (T_reg_), follicular T helper cells (T_fh_), astrocytes, dendritic cells (DCs), and tissue-resident macrophages which establish in different tissues soon after acquiring infection [21,22]. In light that current approaches including ART are unable to prevent acute mucosal CD4^+^ T cell depletion after first exposure to the virus, gaining a better understanding of permissive cell types, mechanisms of transmission, and cellular reservoirs could be beneficial to devise new approaches for HIV eradication efforts. For instance, Th_17_ antiviral function of HIV infection is not completely understood; however, Th_17_ cells are key to HIV pathogenesis and represent potential targets to help prevent or reduce HIV transmission. Here, we review important insights and recent findings of the role of Th_17_ cells during HIV pathogenesis in mucosal surfaces, and the mechanisms of HIV-1 infection spread among T cells in tissues (Figure 1).

## 2. Mucosa HIV-1 Pathogenesis and the Role of Th_17_ Cells

Mucosal surfaces are crucial to HIV-1 transmission, as these constitute the boundary between the host and the environment [21,22]. The mucosal immune system (commonly described as the mucosa-associated lymphoid tissue or MALT) is the largest epithelial surface consisting of lymph nodes (LNs) and lymphoid tissues present in submucosal layers of the gastrointestinal (GI), respiratory, urinary, and genital tracts, in addition to eyes, tonsils, thyroid, breasts and salivary glands [10,23]. The gut-associated lymphoid tissue (GALT) is part of the MALT and includes Peyer’s patches in the small intestine and mesenteric lymph nodes (MLNs) [21,22]. The GI mucosa represents a cardinal site of HIV-1 pathogenesis due to its role as a portal of entry and as a site of infection dissemination to lymphoid tissues [24,25]. During mucosal viral exposure, HIV-1 targets CD4^+^ T cells, macrophages, Hofbauer cells (HCs or placental macrophages), Kupffer cells (KCs or liver macrophages), DCs, DC-SIGN^+^ DCs, Langerhans cells (LCs), and mast cells (MCs) located in the epithelial layer or within the vicinity, which can be responsible for residual replication [9,26,27,28,29,30]. It is well established that HIV-1 induces a progressive and steady loss of CD4^+^ T cell count, leading to the impossibility of containing HIV infection, which is characteristic of HIV pathogenesis and culminates in AIDS progression [9,26,27,28,29,30].

The three recognized stages of HIV infection, acute infection, chronic infection, and AIDS, are defined by viral load, CD4^+^ T cell count, as well as clinical progression; however, prior to the appearance of acute infection symptoms, there is an initial eclipse phase (up to 10 days) in which infection is established at the exposure site, while viral load has not yet reached detectable levels in the circulation [31]. The acute or primary infection phase of HIV is recognized as the time from acquisition until seroconversion (up to 4 weeks from first exposure) [9]. Acute infection is characterized by “flu-like” symptoms along with high levels of viremia (up to 10^7^ or more copies of viral RNA per mL of blood), a substantial drop in both peripheral and lymph nodes CD4^+^ T cell counts, and an increase in overall CD8^+^ T cells [9,31,32]. The chronic phase of infection or clinical latency (1 to 20 years after acute infection) is characterized as an asymptomatic phase with a continued decline of CD4^+^ T cells, usually correlating with AIDS progression along with the level of immune activation, presenting with constant or slow levels of viremia (in the order of 1 to 100,000 copies/mL) [33,34]. AIDS is the final stage of HIV infection, when CD4^+^ T cells abruptly decline (below 200 cells per mm^3^, in contrast to the normal range between 500 and 1500 cells per mm^3^), at this point, HIV-1 infection control is lost, viremia rises, and opportunistic infections also rise as a result of CD4^+^ T cells depletion. This stage often culminates in death [9,33,35]. Mucosal CD4^+^ T cell recovery during chronic infection is often used to predict the clinical outcome with no recovery in rapid progressors, and only transient recovery in normal and long-term progressors [24]. Unfortunately, undetectable plasma viral loads are no guarantee of viral particles’ absence. In fact, HIV RNA and DNA have been identified in GALT despite ART and undetectable viral loads in plasma [36]. Furthermore, most individuals under ART with plasma HIV-1 RNA suppressed below the limits of detection (20 to 40 copies/mL) as per commercial assays, still show detectable HIV-1 RNA in plasma (1 to 3 copies/mL) by RT-qPCR [37].

Prior to dissemination and latency, HIV-1 establishes infection by a single viral particle infecting a single cell, mainly a CD4^+^ T cell [9,38,39]. CD4^+^ T cells represent most cells residing within the GI tract, LNs, and other lymphatic tissues [40]. GALT is recognized as the primary site of HIV replication where CD4^+^ T cells are massively and rapidly depleted during primary infection; however, and contrary to older beliefs, the activation state of target cells is not required for acquiring HIV infection [36,41,42,43]. Macal et al. demonstrated that a good number of CD4^+^ T cells in GALT of HIV-infected patients are not activated [36,41]. HIV-1 entry into target cells requires the engagement of the surface subunit gp120 of the viral envelope glycoprotein (Env) to host CD4, and C-C motif chemokine receptor 5 (CCR5) and/or C-X-C motif chemokine receptor 4 (CXCR4) serving as HIV co-receptors in the host cell membrane [11,44]. Viruses using CCR5 co-receptors (also known as R5 viruses/R5-tropic strains) are responsible for viral transmission and establishment of infection, while viruses using CXCR4 (also known as X4 viruses/X4-tropic strains), or both co-receptors (also R5X4 viruses/dual-tropic strains) have been identified at later time points during disease progression [38]. Both HIV-1 and the lab SIVmac model preferentially infect T cells expressing CCR5 co-receptors [39,45,46]. Indeed, CCR5^+^ CD4^+^ memory T cells have been reported to constitute most CD4^+^ T cells present in MALT as opposed to CCR5^−^ CD4^+^ T cells mostly present in peripheral blood and LNs [33].

Once HIV-1 infects a CD4^+^ T cell, it integrates its genetic material into the host cell’s DNA to either initiate a cycle of replication or remain inactive [47]. After integration, the HIV-1 genome resides within the DNA of the infected CD4^+^ T cell, acquiring lifelong persistence [33,36]. Once a replication cycle is completed, mature HIV-1 virions are released into the extracellular space ready to spread and infect other host-permissive CD4^+^ T cells. CD4^+^ T cells participate in orchestrated cascades of immune responses against acute and chronic viral infections, and they serve as mediators between innate and adaptive immunity [8,34,48]. One peculiarity of CD4^+^ T cells is that they can differentiate into different T cell subsets responsible for mounting specific adaptive immune responses [49]. By the same token, different subsets of CD4^+^ T cells possess different levels of susceptibility and permissiveness to HIV infection [42]. Among CD4^+^ T cells, Th_17_ cells represent a major T cell lineage at mucosal sites known to be highly susceptible and permissive to HIV/SIV entry, and capable of supporting intracellular viral replication [26,33,42,50,51,52]. Th_17_ cells are primarily enriched in the intestinal lamina propria (LP) and vaginal cervix mucosa [33,49,53,54,55].

Th_17_ cells were identified in 2005 [56,57,58,59]. Th_17_ cells (CCR4^+^ CCR6^+^ CXCR3^−^ CD161^+^) derive from CD161^+^ precursors and constitutively express CCR4 and CCR6, but not CXCR3 [59]. Established T cell lineages are commonly characterized through canonical sets of cytokines and transcription factors [60]. For instance, Th_17_ cells are defined by the expression of a transcription factor profile including retinoic acid-related orphan receptor gamma t (RORγt), RAR-related orphan receptor alpha (RORα), and signal transducer and activator of transcription 3 (STAT3), and by the secretion of cytokines such as interleukin (IL)-8, -17A, -17F, -21, -22, -26, as well as tumor necrosis factor alpha (TNFα), and C-C motif chemokine ligand 20 (CCL20, also MIP-3α) [61]. Th_17_ differentiation requires cytokines such as transforming growth factor-beta (TGF-β) and IL-6 [60,62,63]. Low doses of TGF-β, along with IL-6, induce STAT3 and RORγt expression, which promotes Th_17_ development [51,63,64,65]. In contrast, high doses of TGF-β inhibit RORγt while promoting the generation of inducible Treg (iTreg) cells [51,64,65]. IL-21 is a required survival factor involved in Th_17_ expansion [51]. IL-1β contributes to Th_17_ differentiation and expansion [51,60,63]. IL-23 is a requirement for pathogenicity in Th_17_ [51]. Together, IL-1β and IL-23 are vital to complete Th_17_ lineage commitment program by repressing IL-10 and inducing B lymphocyte maturation protein-1 (Blimp-1) expression in Th_17_ cells [51,63].

In contrast to Th_1_ and Th_2_ cells, which are considered as stable lineages, Th_17_ cells are more plastic and less terminally differentiated cells capable to undergo lineage reprogramming and transdifferentiate into Th_1_, Th_2_, T_fh_, or T_reg_ like subsets, particularly under lymphopenic or inflammatory conditions [51,63,64,65,66]. Contingent upon the microenvironment, Th_17_ cells have the potential to acquire new effector features and convert toward other lineage subsets [51,66]. Th_17_ cells generated either in vitro or in vivo can retain the Th_17_ phenotype or can acquire new effector characteristics upon secondary stimulation [65]. Th_17_ cells may shift toward Th_1_Th_17_ cells during autoimmunity, cancer, and infections or toward Th_2_Th_17_ cells during asthma [66]. For instance, Th_17_ cells in the presence of IL-12 or IL-23 and in the absence of TGF-β convert toward Th_1_Th_17_ cells (IFN-γ secreting) [51,65]. Lymphopenic conditions also shift Th_17_ cells toward pathogenic IFN-γ secreting cells in models of diabetes mellitus [65]. And TNF-α has been reported to shift Th_17_ cells toward Th_1_Th_17_ subsets [51]. On the other hand, Th_17_ cells exposed to IL-4 acquire the ability to produce Th_2_ cytokines IL-4 and IL-5 while expressing IL-17A, IL-21, and IL-22 [51]. These Th_2_Th_17_ cells have been detected in the peripheral blood of patients with chronic asthma [51]. In the context of autoimmune diseases or infections, Th_17_ cells may also convert toward T_reg_ cells [66]. Intestinal Th_17_ cells can reprogram toward IL-10 producing T_reg_ cells under pro-inflammatory conditions in the gut involving Aryl hydrocarbon receptor (AHR) and TGF-β signaling or upon high levels of TGF-β and retinoid acid [51,64]. Some reports indicate that Th_17_ cells transdifferentiate into T_reg_ cells naturally during the resolution of inflammation [51]. Finally, Th_17_ cells reprogram toward T_fh_ and contribute to the development of IgA-secreting germinal center B cells [51,66].

The dual activity of Th_17_ cells is extensively documented. Th_17_ cells are beneficial in maintaining mucosal barrier integrity and homeostasis [33,49,51,52,53,54,67,68], while the role of Th_17_ cells in maintaining gut integrity is crucial against fungal and bacterial infections [24,51]. One of the mechanisms of Th_17_ cells to maintain epithelial barrier integrity involves CCR6 expression to be recruited to the skin and to the small intestine in response to CCL20 (the ligand for CCR6) where they stabilize by IL-23 [26,53,59,69]. Mucosal epithelial cells secrete CCL20 in response to inflammatory stimuli, including pro-inflammatory cytokines (i.e., IL-1α and TNFα) and bacteria [69]. Then, Th_17_-derived IL-17 and IFN-γ stimulate keratinocytes and APCs to produce more IL-1α, IL-23, and CCL20, resulting in a feedback loop for keratinocyte proliferation and pro-inflammatory cytokines, production and secretion [26,53,59,69]. Conversely, Th_17_ cells are also inducers of inflammation (by recruiting neutrophils, inducing chemokine expression, and releasing inflammatory cytokines) as well as contributors to autoimmune disorders (i.e., multiple sclerosis, psoriasis, rheumatoid arthritis, inflammatory bowel disease, systemic lupus erythematosus, and asthma) [33,49,51,52,53,54,67]. Furthermore, Th_17_ cells are recognized as vital effector cells in adaptive immunity able to recruit epithelial cells, neutrophils, and B cells, in addition to directly responding to infections by pathogens including fungi (i.e., Candida albicans), mycobacteria (i.e., Mycobacterium tuberculosis), and extracellular bacteria (i.e., Klebsiella pneumoniae) [33,49]. Recent reports by Agak et al. identified a subpopulation of Th_17_ cells capable of capturing and killing extracellular bacteria by secreting antimicrobial proteins and T cell extracellular traps (TETs) in cell culture systems [60].

Observations from several groups suggest that HIV infection of T cells is enhanced under conditions containing Th_17_ polarizing cytokines (i.e., IL-1β, TGF-β, IL-6, and IL-23) in culture systems; furthermore, Th_17_ cells are preferentially depleted from GALT during acute HIV/SIV infection in vivo [52,70]. Current evidence has led to wonder what are the features that make Th_17_ cells susceptible to HIV/SIV infection and depletion. There is still no consensus to answer this question, but several proposals are considered.

## 3. Mechanisms That Contribute to the Preferential Loss of Th_17_ Cells during HIV-1 Infection

The rapid loss of Th_17_ cells is documented to be key to HIV/SIV pathogenesis [33,42,51,52]. Maek et al. pioneered reports highlighting increased IL-17 production by circulating T cells, and the role of Th_17_ cells during HIV infection [71]. It is recognized that Th_17_ cells are susceptible to HIV/SIV entry, then after successful virus internalization, Th_17_ cells can support virus replication and production [52]; however, there is evidence that Th_17_ cells play inhibitory effects against HIV replication and amplification [72]. Mechanisms contributing to the preferential depletion of Th_17_ cells during HIV/SIV infection in vivo remain mainly unknown since most analyses have been generated from culture systems using purified Th_17_ cells from either sorted human peripheral blood or from T cells skewed to Th_17_ [73]. Some of the proposed mechanisms explaining the preferential loss of Th_17_ cells include: (a) overexpression of factors inhibiting Th_17_ differentiation, (b) depletion of naïve Th_17_ precursors, (c) deficient expression of cytokines such as IL-21, (d) high expression of HIV binding receptors, (e) low expression of macrophage inflammatory protein-1β (MIP-1β is an HIV inhibitory chemokine, also known as CCL4), (f) AIDS-associated opportunistic infections, (g) chronic immune activation, (h) low expression of HIV-suppressive RNase 6, (i) susceptibility and permissiveness to HIV/SIV infection, and (j) expression of co-receptors and integrins (such as CD4, CXCR3, CXCR4, CCR4, CCR5, CCR6, and α4β7 integrin) among others [42,51,52,69,74]. Here, we only focus on Th_17_ susceptibility and permissiveness to HIV/SIV infection, and Th_17_ expression of co-receptors and integrins, both of which represent mechanisms targeting Th_17_ cells for preferential infection and depletion.

It is thought that high levels of CCR5 expressed by Th_17_ cells facilitate virus entry [26]. In support of this notion, CCR5^+^ Th_17_ cells are found depleted from the GI tract of HIV-infected patients [51]. Some CCR6^+^ cells have also been documented to express higher levels of CCR5, as compared to CCR5^−^ cells [42]. Further, co-expression of CCR6 and CCR5 in Th_17_ cells correlates to their depletion from the blood of HIV-infected patients [51]. Planas et al. demonstrated that Th_17_ polarized CCR6^+^CD4^+^ T cells are highly permissive to infection [69]. Although the CCR6^+^ CD4 T cell population is inclusive of all Th_17_ cells, not all CCR6^+^ cells are capable of secreting IL-17 [26,69]. Subsets of CD4^+^ T cells are commonly defined by their functional properties [69]. Similarly, signature cytokines production is used to characterize these subsets, and their expression of chemokine receptors are used to identify functionally polarized CD4^+^ T cell subsets such as Th_1_ (CXCR3^+^/CCR4^−^/CCR6^−^), Th_2_ (CXCR3^−^/CCR4^+^/CCR6^−^), Th_17_ (CXCR3^−^/CCR4^+^/ CCR6^+^), and Th_1_/Th_17_ cells (CXCR3^+^/CCR4^−^/CCR6^+^) [19]. Wacleche et al. described four IL-17A producing Th_17_ subsets in humans based on their CCR4 and CXCR3 expression including Th_17_ (CCR4^+^ CXCR3^−^), Th_1_Th_17_ (CCR4^−^ CXCR3^+^), and two Th_17_ polarized subsets designated as CCR6^+^DN (CCR4^−^ CXCR3^−^) and CCR6^+^DP (CCR4^+^ CXCR3^+^) both, in vitro and in ART-treated patients where each subset is thought to play a specific role during HIV pathogenesis [51]. Preferential loss of CCR6^+^ CD161^+^ CD4^+^ T cells from the blood of SIV-infected rhesus macaques (but not in sooty mangabeys, a natural SIV host) correlates to disrupted homeostasis and contributes to disease progression due to redistribution to the gut mucosa [69]. Analyses from a vaginal challenge model in rhesus macaques detected that SIV selectively targeted CCR6^+^ CD4^+^ T cells corresponding to the Th_17_ lineage as confirmed by RORγt expression [26]. Th_17_ cells were highly susceptible to SIV and selectively depleted from the female reproductive tract (FRT) early within 48 h post-infection [26]. Several studies in SIV-infected rhesus macaques indicate that Th_17_ cells are preferentially depleted from the GI tract during the acute phase of infection as compared to blood Th_17_ cells; moreover, Th_17_ cells are preserved during non-pathogenic infection [43,51]. Additionally, a variety of publications indicate preferential depletion of Th_17_ cells from HIV-infected patients. In a cross-sectional study from a South African cohort, Mycobacterium tuberculosis-specific Th_17_ cells were preferentially depleted in HIV-infected patients [74].

In HIV patients with progressive disease, Th_17_ frequency is lower during the chronic phase [43]. Lower frequencies of peripheral blood CCR4^+^ CCR6^+^ Th_17_ and CXCR3^+^ CCR6^+^ Th_1_Th_17_ cells have been reported in chronic HIV-infected patients under ART when compared to uninfected patients [51]. Nevertheless, Th_17_ cells are preserved under slow disease progression or during non-pathogenic infection [43,75]. For instance, sooty mangabeys which do not progress to AIDS preserve healthy mucosal function as well as Th_17_ levels post-SIV infection [42]. In contrast, in HIV-infected long-term non-progressors, the frequency of Th_17_ cells is preserved [51]. While the frequency of Th_17_ cells negatively correlates with plasma viral load, it positively correlates with CD4^+^ T cell counts [43,51,76]. Assessments regarding Th_17_ frequency may vary in the literature due to the use of diverse methods of characterization and identification (i.e., surface markers vs. intracellular production of cytokines) [73]. In summary, Th_17_ depletion induce enhanced mucosal permeability and bacterial translocation leading to chronic immune activation (driver of changes in the frequency of different T cell subsets such as an increase in effector or fully differentiated T cells and a decrease in naïve T cells) and AIDS progression [43,51,76]. Depletion of Th_17_ cells in the blood and gut has been identified in both humans and macaques with HIV or SIV infection, and it is a predictor of disease progression [75,77]; however, blood Th_17_ depletion analyses are still filled with questions, as Th_17_ cells function mainly at the mucosa and not in circulation [77].

Th_17_ and T_reg_ subsets derive from a common progenitor and differentiate based on IL-6 and TGF-β levels; however, they have quite opposite functions, and their ratios are directly associated with HIV progression [43,72,78]. While Th_17_ cells’ primary function is to mount immune responses to invading pathogens via pro-inflammatory responses and perhaps promote autoimmunity, T_reg_ cells have an immunosuppressive function and help maintain self-tolerance, control activation, and expansion of autoreactive CD4^+^ T effector cells via anti-inflammatory responses [43,72,78]. Normally, Th_17_/T_reg_ ratios are stable; however, inflammation and other immune conditions, including multiple sclerosis, rheumatoid arthritis, inflammatory bowel disease, and HIV/SIV infections drive generalized immune activation and disturb their balance [72,79]. During acute HIV infection, T_reg_ cells may be beneficial to the host, as prior to full activation of HIV-specific immune responses, T_reg_ cells inhibit T cell activation and limit the number of target cells for HIV spread [80]; however, during chronic HIV infection, increased T_reg_ cell frequency is detrimental to antiviral immune responses [80]. Changes in the absolute numbers of T_reg_ and Th_17_ cells lead to imbalanced Th_17_/T_reg_ ratios, which contribute to the breakdown of mucosal integrity, resulting in microbial translocation and systemic immune activation [79]. Falivene et al. found reduced Th_17_/T_reg_ ratio in HIV-infected patients as compared to healthy donors, and higher Th_17_ levels correlating with stronger CD8^+^ T cell responses against the infection which led them to suggest that Th_17_ cells have potential prognostic value for HIV-specific T cell responses [77]. Thus, a progressive increase in T_reg_ frequency along with a progressive loss of Th_17_ drive Th_17_/T_reg_ ratios to drop as HIV infection progresses [72,80]. Moreover, high frequency of both Th_17_ and Th_17_/T_reg_ ratios are reported in HIV elite controllers when compared to HIV patients [43]. To date, we have a better understanding of the role of Th_17_ cells in HIV-1 infection control; however, the main mechanisms of HIV-1 transmission in Th_17_ cells remain unclear. Hot topics of research include whether HIV-1 spread takes place mainly via cell-free viral particles and/or via cell-to-cell direct contact [81].

## 4. Cell-to-Cell versus Cell-Free HIV-1 Spread

After the assembly of infectious virus particles, HIV-1 is proposed to infect and replicate in target cells via multiple mechanisms [82]. The two main modes of HIV-1 spread among permissive cells are direct cell-to-cell infection and cell-free infection [82,83,84,85,86]. Although the release of cell-free viral particles has been considered as the primary mode of HIV-1 infection transmission, cell-to-cell and cell-free modes are not mutually exclusive and the precise contribution of either mode of virus transmission in vivo is not yet clear [87,88]. HIV-1 Env spike supports both cell-free and cell-to-cell infection of CD4^+^ T cells [44]. HIV-1 Env spike is a trimeric glycoprotein comprised of three gp120-gp41 heterodimers which mediate viral attachment, fusion, and entry into CD4^+^ T cells during cell-free and/or direct cell-to-cell infection [44,89]; however, it has been proposed that only cell-to-cell HIV-1 transmission can overcome deficiencies of viral Env incorporation [44].

Direct transfer between one donor cell and a target cell by cell-to-cell spread has been extensively characterized in cell culture systems using T cells from peripheral blood lymphocytes [11,82]. The first description of direct cell-to-cell HIV transfer was reported between DCs and T cells [90]. Nevertheless, cell-to-cell HIV transfer takes place between a number of immune cells such as macrophages and LCs, which are known to help establish HIV reservoirs in different host tissues and play important roles early during transmission and dissemination [81]. Direct contact between infected and uninfected cells contributes to viral spread through well-described structures such as the virological synapse (or VS, described as interactions engaging the Env glycoproteins expressed in the infected cells and receptor in the target cells), filipodia, and nanotubes in addition to phagocytosis, and cell-cell fusion modes of transmission [11,81,85,90]. Galloway et al. proposed that infected cells in lymphoid tissues are the main source of HIV spread via direct cell-to-cell infection [83]. It is speculated that infected T cells in LNs possessing migratory potential contribute to cell-to-cell transmission and spread in vivo [81]; however, HIV transmission either by cell-to-cell or by cell-free modes has been commonly assessed in vitro [82]. The challenge to quantitatively discriminate the effectiveness of each HIV-1 transmission strategy individually rests on technical difficulties to exclusively analyze cell-free infection without cell-to-cell infection taking place in parallel, and vice versa, since these are not mutually exclusive mechanisms [82].

Cell-to-cell transmission has been regarded as an efficient strategy implicated in HIV-1 pathogenesis [83,85]. Multiple studies have suggested that a virus associated with a cell is more infectious than a cell-free virus [91]. Experimental and mathematical models allowed for the quantification of the sole dynamics behind cell-to-cell infection, leading to the finding that cell-to-cell infection predominates 60% of total viral infection [84]. Chen et al. reported that in vitro cell-associated infection is 18,000-fold more efficient in transferring viral particles into target cells than cell-free infection [90]. Cell-to-cell transmission has been found to reduce the generation time of viruses by 0.9 times while increasing viral fitness by 3.9 times [84]. Furthermore, it has been argued that cell-to-cell contact through the VS may protect HIV-1 from antiviral factors such as antibodies while also enabling disseminating [90,91]. Numerous studies indicate that gp120-directed and gp41-directed broadly neutralizing antibodies halt virus transmission in rhesus macaques upon topical or intravenous challenge of cell-free virus [92,93]. Although spread via cell-free particles has been challenged as the main transmission model, there is evidence in the literature for its support. After comparing HIV and human T lymphotropic virus (HTLV) transmission, HIV spread was proposed to take place mainly via cell-free mode [94]. A big challenge to cell-free HIV-1 transmission implies that during transcytosis via mucosal epithelial cells, only 0.01% to 0.05% of virions from the initial inoculum may translocate across epithelial cells [95,96]. It is documented that more than 90% of virions internalized in tonsil, cervical and foreskin epithelial cells do not cross the epithelium; instead, virions are retained in endosomal compartments, such as multivesicular bodies and vacuoles for several days [96]. Sequestered virions in the epithelium maintain infectivity for about nine days and can be released through cell-to-cell interaction of epithelial cells containing the virus with activated peripheral blood mononuclear cells (PBMCs) and CD4^+^ T lymphocytes [96]; however, Sufiawati et al. reported that HIV-1 cell-free virions along with tat and gp120 proteins are key for the disruption of adherens and tight junction proteins leading to the impaired mucosal barrier and spread of the virus within target cells, as seen in ex vivo tonsil tissue explants [95]. In addition, HIV-1 and human cytomegalovirus (HCMV) coinfection of tonsils act synergistically to promote and facilitate both viral infections [95].

The infectivity of retroviruses, including HIV-1, either in plasma or cultured media, has been reported to be less than 0.1%, arguing potentially large numbers of defective virions in a virus pool and limiting the infectivity of HIV virions [97]. Nonetheless, Josefsson et al. demonstrated that in peripheral blood CD4^+^ T cells from patients, the majority of infected cells contain only one copy of HIV-1 DNA as compared to the high proviral HIV content present in tissues and co-cultures which may correlate to transmission by cell-free particles [98,99]. Despite the fact that HIV-1 infects antigen-presenting cells (APCs) to a lesser extent than T cells, a major pathogenic process in HIV-1 infection is the uptake of HIV-1 by APCs followed by transfer of virus to CD4^+^ T cells, leading to explosive levels of virus replication within T cells [100]. Nonetheless, DCs express both CD4 and CCR5 HIV-1 co-receptors where DC-mediated cell-free HIV-1 trans-infection of T cells is well documented [101]. Additionally, three nonexclusive pathways are described for cell-free viruses to enter DCs including clathrin-rich endosomes, lipid rafts in the presence of DC-SIGN (pathway which favors productive infection), or via lipid rafts in the absence of DC-SIGN (pathway which may prevent viral replication) [101].

The debate on which mode of transmission is more efficient is still ongoing, with no concrete conclusion yet. Most of the available evidence has been obtained from in vitro systems examining virus transmission; however, the actual interactions and conditions in vivo are not fully understood. Notwithstanding, infected breast milk, semen, and vaginal secretions contain a mix of both cell-free viral particles and infected cells [82,93]. Current evidence indicates that the field is not plain when making comparisons between cell-to-cell and cell-free HIV-1 transmission modes, as cell-free viral particles might also contribute to cell-associated infection. HIV likely takes advantage of both modes of transmission to spread. Moreover, no single cell type has been linked to a preferential mode of transmission, and it is widely unknown what the preferential mode of transmission in cells typically found in the mucosae such as Th_17_ lymphocytes is. The next difficult question is how latent HIV-1 reservoirs replenish?

## 5. Virus Free Seeding New Reservoirs in Distant Places and Latency

ART regimens are unable to halt chronic immune activation, inflammation, and immune dysfunction, all of which may contribute to the establishment of reservoirs harboring latent HIV-1 [102]. Thus, it is likely that the cure for HIV-1 is a reservoir away. Previously, the rapid loss of CD4^+^ T cells was associated with apoptosis, but it was found that about 95% of quiescent CD4^+^ T cells die via caspase 1-mediated pyroptosis and which correlates with chronic inflammation in HIV pathogenesis [103]. Later, Galloway et al. went on to show in cell culture that, unlike cell-free HIV-1 particles, cell-to-cell contact predominantly through the VS is key to trigger innate immune responses leading to the depletion of non-permissive CD4^+^ T cells via caspase 1-dependent pyroptosis [83]. In contrast, highly permissive cells in HIV-1 infection, such as activated peripheral blood lymphocytes die via caspase 3-mediated apoptosis [83]; however, not all infected cells die, and many HIV-infected cells remain as a latent reservoir. Memory CD4^+^ T cells represent the primary HIV reservoir in tissues [104]. The seeding of free HIV virions to distant places is subject to the limits of diffusion between tissues and restricted by not just anatomical barriers, but by soluble immune factors as well (i.e., complement factors and antibodies) [105]. In contrast, cell-to-cell transmission in LNs with high local density of target cells may contribute to the spread of infected cells given the migratory potential of T cells which might transport HIV to remote tissues [105]. Moreover, cell-to-cell transfer represents a mechanism for HIV to hide and escape from the immune system and ART, contributing to the establishment of new virus reservoirs and latency in distant host tissues [81]. Thus, the cell-to-cell spread of HIV is likely promoted in lymphoid tissues with a high abundance of target cells in proximity, along with reduced ART penetration [106].

Data from latent reservoirs in patients on ART exhibit a very slow decay rate (t_1/2_ = 3.7 years) which translates to about 73 years to eradicate a reservoir of 10^6^ cells, making cure unlikely even under ART [18]. Additionally, infected CD4^+^ T cells are long-lived cells capable of living for decades; however, these can potentially revert to resting memory CD4^+^ T cells and further contribute to latent HIV reservoirs [18,36]. CCR5^+^ CD4^+^ memory T cells constitute most CD4^+^ T cells in MALT as opposed to CCR5^−^ T cells mostly present in peripheral blood and LNs [33]. Most CCR5^+^ CD4^+^ memory T cells are preferentially depleted during early HIV/SIV infection [70]. In fact, memory CD4^+^ T cells have been reported to be more permissive to HIV compared to naïve T cells [70]. CCR5^+^ CD4^+^ memory T cells have been identified as specific targets of HIV replication and infection [33]. Monteiro et al. suggested CCR6 as a marker for memory T cells imprinted with a transcriptional program permissive to HIV replication [70]. Importantly, CCR6^+^ T cells also express integrin β7 and CCR5, which possess superior capabilities to disseminate HIV from the entry site since integrin β7 can regulate cell migration into the GALT and bind to HIV-gp120 [70].

Studies by Meås et al. using a cell-to-cell transmission model show that toll-like receptor 8 (TLR8) activates human T cells and triggers inflammatory responses favoring both HIV-1 replication and reversal of latency [11]. Reversion of latency was also observed in patient-derived latently infected CD4^+^ T cells by TLR8 stimulation [11]. Moreover, TLR8 stimulation promoted differentiation towards pro-inflammatory Th_17_ cells by upregulating IL-17 production [11]. Furthermore, Hsiao et al. found that CCR5-tropic HIV could not enter naïve CD4^+^ T cells but gained entry to all subsets of memory CD4^+^ T cells including tonsillar memory cells expressing the IL-7 receptor alpha chain or CD127^+^ tissue memory (TM) cells that preferentially support latent HIV-1 infection as demonstrated by HIV DNA integration but not HIV gene expression [104]. The authors propose that these CD127^+^ TM cells represent a superior alternative to in vitro tissue models of HIV latency based on blood-derived cells which present with early post-entry by SAM domain and HD domain-containing protein 1 (SAMHD1) restriction [104]. Moreover, T cell latent HIV-1 reservoir include infected cells in diverse locations including, but not limited to peripheral blood, LNs, central nervous system (CNS), GALT and tissues such as lungs.

## 6. Conclusions

There are multiple challenges in the field. First, we know that ART does not cure HIV and that HIV reservoirs and latent infection are in part to blame. Moreover, HIV can accumulate genetic diversity over time during infection on a given individual, making the clearance of the virus on an infected patient unfeasible [9]. Another challenge is the absence of a good animal model for HIV. Commonly, non-human primates and SIV or chimera simian/HIV (SHIV) are used, but they differ from HIV-1. Alternatively, humanized mice (mice with a reconstituted human immune system) allow for features that correlate better with HIV-1 transmission in humans [107]. Furthermore, tissue environments may have a direct impact on virus spread, as well as the modes of viral transmission; however, this remains to be established [88]. A better understanding of the mechanisms promoting and supporting latency in tissue cells could help devise innovative approaches to identify and eliminate latent reservoirs of infected cells in order to devise new strategies to cure HIV/AIDS [104].

## Figures and Tables

**Figure 1 viruses-14-00404-f001:**
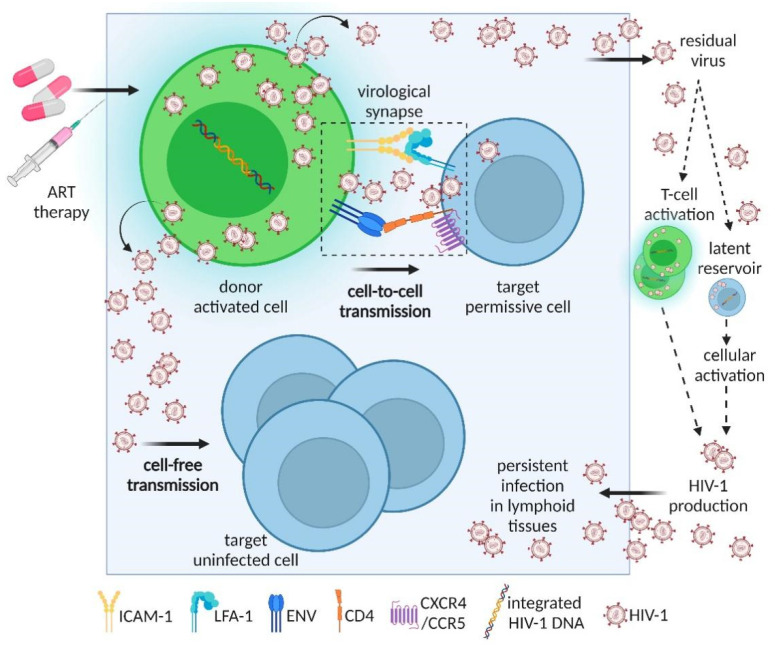
Overview of current models of HIV-1 transmission between CD4^+^ T cells.

## Data Availability

Not applicable.

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
