# Peer review of "HIV Infection and Spread between Th17 Cells"

_viruses, 2022, doi:10.3390/v14020404_

Round 1

Reviewer 1 Report

I believe that the article discusses a timely topic with good expertise regarding the subject matter. The topic discussed is an important aspect of immunology that warrants extensive review. The subject matter is impactful, clinically relevant, and an expanding subject of biomedical research. Past and future studies on the subject matter are and will continue to be high impact research that requires a great deal of creativity to address. I applaud the authors for reviewing the subject matter and doing so in a succinct and well curated manner. The figure for the manuscript is also quite impressive.

The review is well written but possess minor grammatical issues throughout. However, the formatting of the references section is not currently acceptable (see major revision 1). While the text itself only needs minor revisions primarily, I believe the formatting of the references needs to be corrected prior to my acceptance.

I believe that after correction of these issues the article will be a great manuscript that warrants publication within the journal “Viruses”.

Minor Revisions

  1. Line 35: Single-stranded RNA is mentioned at line 35 prior to the acronym definition at line 41. Please define “ssRNA” at line 35 and refer to it as such throughout.
  2. Line 47: Please rephrase or break up the sentence at line 47 for clarity.
  3. Line 55: Please consider using “unable to” instead of “incapable to” for clarity purposes.
  4. Line 57: The ending of the sentence with “and efficiently suppress new” does not make sense. I believe there was meant to be more here.
  5. Line 58: I believe it would be appropriate to briefly mention approaches that aim to address the issue of HIV reservoirs, i.e shock & kill techniques and deep latency techniques
  6. Line 64: I believe this should read “establish” instead of “establishing”
  7. Line 68: Just “reservoirs” seems out of place when listed in this sentence. Consider expanding what in particular about HIV reservoirs.
  8. Line 68: The sentence ending on this line should have an accompanying reference.
  9. Line 76: The sentence beginning at this line needs to be adjusted. It is not clear that only some lymph nodes are part of mucosal immunity, such as the mesenteric, while most would be a part of the peripheral immune system. I also don’t agree with the mucosal associated lymphoid tissues being non-organized.
  10. Line 84: The sentence beginning at this line feels out of place.
  11. Line 101: throughout mm3 not mm3, please fix this here and throughout
  12. Line 118: Should read “are not activated”
  13. Line 139: Express is misleading in this context when regarding T cells, consider using “possess” or a similar term instead.
  14. Line 152: I believe it should be “promotes” and not “promote” at this line.
  15. Line 154: For the sentence ending at this line please include a citation.
  16. Line 157: Break up these citations so they accompany the statements that belong to in the preceding sentences from lines 154-156. This manner of citing only a concluding remark instead of the supporting statements is confusing to the reader and should be avoided.
  17. Line 159: I do not find that this citation sufficient for the statement. The article only discusses Th1-Th17 switching. It appears the cited article briefly mentions and cites articles pertaining to the transition between Tregs-Th17 cells at sites of inflammation for instance, please cite those articles directly and ensure there is supporting citations for all T cell subsets listed.

    Additionally, I was under the impression that under biological conditions these cells differentiate into Th1 or Treg like Th17 cells, and not into proper Th1 cells or Tregs? I believe that they still retain certain characteristics of Th17 cells despite their change in phenotype, can you clarify this for me or adjust the statement in the manuscript to reflect this? It is entirely possible I am mistaken by this as T cell biology is a constantly sliding greyscale when defining cell types.
  18. Line 164: I believe it should read “involves” instead of “involve” at this line
  19. Line 166: For the sentence ending on this line please expand a bit upon what Th17 cells do after chemotaxis to barrier sites.
  20. Line 167: this should read “by recruiting neutrophils, inducing chemokine expression, and releasing cytokines”
  21. Line 171: For clarity purposes please remove “, not just …. B cells, but” and rephrase the remaining sentence.
  22. Line 221: There should be no “,” after rhesus macaques if I am reading this sentence correctly.
  23. Line 224: Please replace “hrs.” with “hours” and be certain to remove the period.
  24. Line 225: please remove the comma between macaques and indicate
  25. Line 228: Please either break this into two sentences with the first ending after “leading to chronic immune activation” or rephrase the current sentence, as it reads like immune activation is the sole factor of driving AIDs development following HIV infection. Some more context of why the immune activation at this site worsens symptomology in HIV infection would be beneficial to the readers. The comma before “if untreated” is also unnecessary at the moment so be sure to fix that following adjustment.
  26. Line 233: A lot has been discussed in this preceding paragraph. Please add a sentence or two summarizing what was discussed. It would be good to do this for all paragraphs of this nature throughout the document. It will aid readers who are new to the subject which are often the target audience of review papers.
  27. Line 234: This should read “Depletion of Th17 cells in the blood and gut has been identified”
  28. Line 235: replace “, respectively” with “and”
  29. Line 236: I believe this should read “at the mucosa and not in circulation”
  30. Line 256: Please remove discussion on HIV/SIV infection from this sentence. You discuss it in more detail further on. In this sentence, please stick to how Th17/Treg ratio goes up in autoimmunity, and then continue on to discuss how the ratio goes down in HIV/SIV infection as you have in the following sentences.
  31. Line 259: Move the sentence regarding elite controllers downward (i.e after the sentence that ends on line 264) so you discuss the normal pathology of Th17/Treg imbalance before discussing the rare exceptions.
  32. Line 285: Dendritic cells are mentioned at line 283. Please move the acronym definition to this point and refer to them as DCs throughout the manuscript afterwards.
  33. Line 290: Remove “, reviewed elsewhere”
  34. Line 295: Quantitatively, not quantitively
  35. Line 296: I believe this should be “rests on”, not “rest on”.
  36. Line 297: remove “while trying”
  37. Line 301: I believe this should read “allowed for the quantification of the sole dynamics behind cell-to-cell infection, leading to … “
  38. Line 303: I believe this should read “predominates 60% of total viral infection”, however I may be reading this part wrong
  39. Line 308: virological synapses were mentioned at line 288, please adjust the acronym placing accordingly
  40. Line 313: HTLV has not been defined in the document and I believe is not discussed elsewhere, please change to human t-lymphotropic virus.
  41. Line 315: add % after 0.01
  42. Line 324: I believe this should read “adherens and tight junction proteins”
  43. Line 326: similarly to HTLV at line 313, expand human cytomegalovirus as HCMV was not discussed sooner.
  44. Line 328: I believe there should be a comma after “cultured media”
  45. Line 329: I believe there should be a comma after “0.1%”
  46. Line 336: I believe there should be a comma after “CD4+ T cells”
  47. Line 348: indicates, not indicate
  48. Line 393: remove the comma after “both”
  49. Section 4 Cell-to-cell versus cell-free HIV-1 spread: Is there no data regarding the methods of HIV transmission pertaining specifically to Th17 cells that differs in other CD4 T cell subsets? While the mechanisms of transmission and the role of Th17 cells was discussed, I was hoping there would be more specific discussion regarding mechanisms / and resultant effects specific to Th17 cell transmission and how these differ from the infection of other CD4+ T cell subsets in this section. I believe this would greatly help both the flow of the manuscript as well as the impact of it to add a short discussion on the topic.

Major Revisions

  1. Reference section: Many of the references do not list the journal they are published in (i.e references 3, 4, 5, 7, and many more). Please correct this and make the referencing style uniform. This is a major issue.

Author Response

Reply attached as docx file

Reviewer 2 Report

This article describes extensively, not just Th17, T cell targets of hiv-1 infection. The review is well written and easy to follow for a broad audience. 
I have a few minor comments:
- Perhaps the cells that are responsible for residual replication and the anatomical sites involved should be discussed (lines 110 - 111).
- There should be verification and consistency between "single particle viral" infection (lines 112 - 113) and cell-to-cell transmission (lines 299 - 301)
- It should be indicated whether the cells described in line 218 are Th17
- In the line Th1Th17 are cited. This population is not defined.
- The Th17/Treg ratio is not described during acute infection.

Author Response

replies attached as .docx.

Round 2

Reviewer 1 Report

I believe all of the issues I had have since been corrected.